# Large Learning Rates without the Agonizing Pain: Dispelling the Curse of Singularities in Deep Neural Networks

## Abstract

Employing large learning rates (LRs) in deep learning can accelerate convergence and improve generalization, but it can also cause training instability and loss explosion: determining an appropriate LR is an often laborious and painful art. Our study into the fine-grained behaviors of parametric singularities, specifically the stable ranks of weight matrices of network components, reveals a strong connection between these singularities and training instability. As training progresses, parametric singularities trend upward, a phenomenon that is directly aggravated by large LRs. Crucially, several training steps before prominent instabilities such as gradient explosions, we observe unusually high parametric singularities across the network components, leading to rank-deficient representations. These representations, in turn, amplify parametric singularities during backpropagation, creating a vicious cycle that eventually results in loss explosions. We refer to this phenomenon as *the curse of singularities*. Building on this understanding, we propose a lightweight and robust stabilization method called Parametric Singularity Smoothing (PSS), which allows for early intervention and mitigates impending instability by smoothing the singular spectra of weight matrices, thereby preventing the curse of singularities. This approach is easy to implement, works at any stage of training by restoring stable training even after instability, has neglectable computational overhead, and, most importantly, frees us from the painful LR fine-tunings to avoid instabilities. Experimental results across various datasets, networks, and optimizers demonstrate that our approach allows a 5-10$\times$ increase in LR without producing instability, attaining better training efficiency and generalization. We release our code for everyone to use our methods and reproduce the experiments, available at `https://anonymous.4open.science/r/ICLR_stability-C69C`.

## 1 Introduction

Applying large learning rates (LRs) in deep learning is widely known to have benefits, e.g., escaping local minima (Li et al., 2019; Jastrzebski et al., 2020), accelerated convergence (Smith, 2017; Smith & Topin, 2019), and better generalization (Lu et al., 2023). However, large LRs also increase the probability of training instability (Lewkowycz et al., 2020; Cohen et al., 2021; Wang & Roberts, 2023), which often leads to loss explosions that waste computational resources, a serious issue in large models. Although recent work has investigated the properties of large LRs from various aspects, there still lacks an effective solution to address the instabilities that are more prominent with large LRs. Nowadays, fine-tuning LRs for stabilities remains commonly indispensable and can be painful, especially for real-world foundation models. For instance, it takes multiple runs to determine a proper LR free of instabilities when training OPT-175B (Zhang et al., 2022), each consuming dozens of days on thousands of GPUs. Our goal is to enable a reliable, stable, automated, restart-free approach to using large LRs.

We start by investigating the behaviors of the stable ranks (Rudelson & Vershynin, 2007), i.e., parametric singularities, of weight matrices across model components, such as the key, query, and value matrices of self-attention heads in transformers (Dong et al., 2021). Recent studies (Mousavi-Hosseini et al., 2022; Zangrando et al., 2024; Feng et al., 2022; Huh et al., 2021; Arora et al., 2019)

have revealed the low-rank properties of parametric matrices in different neural networks. These matrices tend to be increasingly "singular" as training progresses. Such singularities are usually considered to be linked with feature spaces of decreased rankings (Dong et al., 2021; Noci et al., 2022); pronounced parametric singularities generally imply that a few singular vectors dominate the feature space, leading to rank-deficient representations with poor expressiveness (Dong et al., 2021; Noci et al., 2022). Previous work did not discuss or investigate the relationship between singularities and training instabilities; we are the first to study how the network's fine-grained, dynamic singularity behaviors cause loss explosions that prevent further learning, which happens quite commonly for large LRs.

Specifically, we investigate the parametric singularity distributions across network components of training sessions with exploding losses. We find that in practice parametric singularities increase monotonically as training proceeds, with deep layers having higher singularities than shallow layers. Notably, growing singularity trends can be observed across varying LR values and are more pronounced with large LRs. Inspecting the few training steps preceding loss explosions reveals a particular step in which growing singularities in the parametric space yield rank-deficient representations with unusually high similarity in the feature spaces. Specifically, a few dominant singular values in the parametric space overshadow the contributions of other singular directions, particularly in deep layers, leading to a sharp reduction in representational diversity. Updating such networks will further increase the parametric singularities, leading to yet more severe rank deficiencies in the next training step. Within a few steps, this vicious cycle produces loss explosion and puts an end to productive learning. We call this phenomenon *the curse of singularities*, and argue that it is a primary cause of training instability.

Based on these findings, we propose a computationally efficient and robust method called Parametric Singularity Smoothing (PSS) for using large LRs without training instability. PSS enables early detection of instability and restores stability by smoothing the singular spectrums of each weight matrix across the network components, raising stable ranks and avoiding the curse of singularities. This method is simple, easy to implement, and does not require restarts or manual LR tuning when countering training instabilities. It enables the use of large LRs without producing loss explosions, and it can even restore stability after instability has occurred. We evaluated this method using various datasets and networks and found that it increases usable LR by 5-10×, providing potentially better training efficiency and generalized performance.

This work's contributions follow:

- It is the first to describe the dynamic patterns of network singularities and reveal their tight associations with loss explosion, a phenomenon we call *the curse of singualrites*.

- It describes a lightweight, easy to implement and robust method enabling large LRs without loss explosion that does not depend on LR tuning or training restarts.

- The PSS method significantly increases the maximum stable LR by 5-10×, providing potentially better training efficiency and generalized performance.

## 2 ANALYSIS

### 2.1 STABLE RANK AND STABLE JACOBIAN ENERGY

Define a feed-forward neural network as $f(\mathbf{x}; \boldsymbol{\theta}) = f_L(f_{L-1}(\ldots f_1(\mathbf{x}; \mathcal{W}^1); \ldots; \mathcal{W}^{L-1}); \mathcal{W}^L)$ where $\mathbf{x}$ and $\boldsymbol{\theta}$ denote the input and learnable parameters, respectively, $L$ is the number of layers, $f_l$ and $\mathcal{W}^l$ denote the function and the collection of weight matrices of the $l$-th layer. Let $\mathcal{W}^l = \{\mathbf{W}_i^l\}_{i=1}^{k^l}$ where $\mathbf{W}_i^l$ denotes the $i$-th weight matrix of the $l$-th layer and $k^l$ denotes the total number of weight matrices in the $l$-th layer. We use the stable rank (Rudelson & Vershynin, 2007) to represent the degree of singularity of a weight matrix $\mathbf{W}$.

Formally, for a matrix $\mathbf{W} \in \mathbb{R}^{m \times n}$ with rank $r \leq \min(m, n)$, we perform the Singular Value Decomposition (SVD) to obtain its singular values $\{\sigma_i\}_{i=1}^{\min(m,n)}$, left singular vectors $\{\mathbf{u}_i\}_{i=1}^{m}$ where $\mathbf{u}_i \in \mathbb{R}^m$, and right singular vectors $\{\mathbf{v}_i\}_{i=1}^{n}$ where $\mathbf{v}_i \in \mathbb{R}^n$, such that $\mathbf{W} = \sum_{i=1}^{r} \sigma_i \mathbf{u}_i \mathbf{v}_i^\top$. Throughout the paper, we assume all singular values are sorted in descending order, namely $\sigma_1 \geq$

$\sigma_2 \geq \ldots \sigma_r > 0$, and we denote the $i$-th singular value of $\mathbf{W}$ as $\sigma_i(\mathbf{W})$. The stable rank (SR) is defined as follows.

**Definition 1.** *(Stable Rank). The stable rank of a matrix $\mathbf{W}$ is defined as the squared ratios of its Frobenius norm $\|\mathbf{W}\|_F^2$ to its matrix 2-norm $\|\mathbf{W}\|_2^2$, which can be written as*

$$\mathrm{SR}(\mathbf{W}) = \frac{\|\mathbf{W}\|_F^2}{\|\mathbf{W}\|_2^2} = \frac{\sum_{i=1}^r \sigma_i^2(\mathbf{W})}{\sigma_1^2(\mathbf{W})}. \tag{1}$$

SR is a "softer" version of the matrix rank and possesses several desirable properties. It better numerically reflects the singular degree of a weight matrix and while its insensitivity to matrix scales and small perturbations enables use across varying parametric spaces. We therefore use it as a measure of the singularities of weight matrices.

A small SR value intuitively indicates the emergence of pronounced singularities, where a few singular values dominate the others in the weight matrix. These overly dominant singular values can lead to rank-deficient representations, which in turn produce Jacobian matrices that, upon updates, further exacerbate the weight singularities and reduce SR values even more.

We define a new and intuitive descriptor to quantify how the gradient updates will change the parametric singularities.

Formally, for a matrix $\mathbf{W} \in \mathbb{R}^{m \times n}$, let $\mathbf{J_W}(\mathbf{x}) = \frac{\partial \mathcal{L}(\mathbf{x})}{\partial \mathbf{W}} \in \mathbb{R}^{m \times n}$ be the Jacobian matrix of the loss function $\mathcal{L}$ w.r.t. $\mathbf{W}$ computed on $\mathbf{x}$, and we will write $\mathbf{J_W}$ for simplicity. For a singular value $\sigma_i(\mathbf{W})$ with corresponding singular vectors $\mathbf{u}_i$ and $\mathbf{v}_i$, we could project the Jacobian matrix along $\mathbf{u}_i$ and $\mathbf{v}_i$ to find its degree of "energy" projected on this singular component, which is defined as

$$\varphi_i(\mathbf{W}) = (\mathbf{u}_i^\top \mathbf{J_W} \mathbf{v}_i)^2. \tag{2}$$

$\varphi_i(\mathbf{W})$ can be interpreted as the length of the projection of $\mathbf{J_W}$ along the singular vector direction of $\sigma_i(\mathbf{W})$. In general, the higher $\varphi_i(\mathbf{W})$ is, the more energy of $\mathbf{J_W}$ is distributed along the singular vector direction of $\sigma_i(\mathbf{W})$, indicating more substantial parametric updates along that singular component. To capture this, we introduce the concept of Stable Jacobian Energy (SJE), which quantifies the proportion of Jacobian energy concentrated on the top singular vectors within the SR.

**Definition 2.** *(Stable Jacobian Energy). The Stable Jacobian Energy (SJE) of a matrix $\mathbf{W}$ is the ratio of its Jacobian matrix $\mathbf{J_W}$'s energy projected on the singular vectors within the SR to the energy projected on all singular vectors, which is defined as*

$$\mathrm{SJE}(\mathbf{W}) = \frac{\sum_{i=1}^{\lfloor \mathrm{SR}(\mathbf{W}) \rfloor} \varphi_i(\mathbf{W})}{\sum_{j=1}^{\min(m,n)} \varphi_j(\mathbf{W})} \tag{3}$$

*Where $\lfloor \mathrm{SR}(\mathbf{W}) \rfloor$ denotes the floor of $\mathrm{SR}(\mathbf{W})$.*

A large SJE value typically indicates that those top singular vectors within the SR will receive high Jacobian energy, and thus, the parametric updates along those singular components can be larger.

SR and SJE are the two key measures used to describe the behaviors of fine-grained network singularities in this work, and we present the empirical observations in the next section.

### 2.2 THE CURSE OF SINGULARITIES: EMPIRICAL EVIDENCE

We start by illustrating the overall trend of parametric singularities across the training session and discuss the role of LRs during the process. We then illustrate the cyclic interaction between parametric singularities and SJE, investigating network behaviors in the steps leading up to prominent training instabilities, ultimately revealing the existence of *the curse of singularities*. Without loss of generality, we plot figures in this section using data collected from a BERT-base model trained on Wikitext dataset, while the observations are prevalent across networks and datasets.

**Parametric singularities and LR impacts**. In Fig. 1, we present the layer-wise average parametric singularities, measured by SR, across the weight matrices of the query modules $\mathbf{W}_Q$ and feed-forward networks (FFN) $\mathbf{W}_{fc}$ as training advances. The SR values remain stable during the warm-up phase of around 1000 steps, with weights matrices consisting of significant singular values

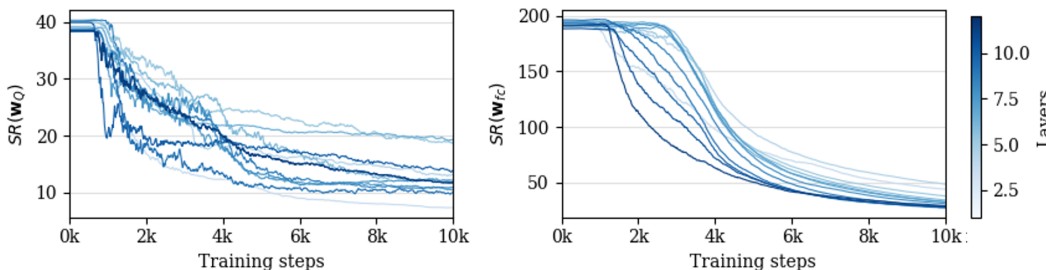

Figure 1: SR curves of 12 layers across training steps. The left and right columns demonstrate the average SRs computed on weight matrices of the query modules and FFNs, respectively. All components of the model share the same trend (See in Appendix A).

on more than 35 singular directions in $\mathbf{W}_Q$ or 180 in $\mathbf{W}_{fc}$. We can reasonably assume that the expressivities of the network at this stage are intact. After the warm-up phase, we could observe that the SR curves demonstrate a continuous downward trend despite some differences in layers and network components. This trend can be explained as when the network learns to fit the dataset, most irrelevant singular components are gradually discarded, and weight matrices focus on several dominating singular directions. Notably, the monotonic increasing trend of singularities is more pronounced in deep layers. We believe that the increased singularities could somehow impair the expressivity capabilities and lead to rank-deficiency representations, which is in accordance with the insights of previous works (Bao et al., 2024).

We further discuss the impacts of LRs on parametric singularities. As shown in Fig. 2(a), a larger LR will generally lead to SR curves of lower values (only an FFN of one layer is shown but this is a common observation). That is, large LRs will typically amplify the effects of network singularities, which are potentially associated with higher probabilities of training instabilities.

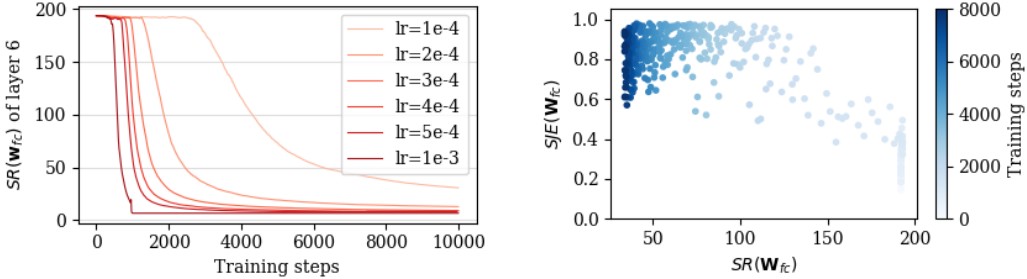

Figure 2: a) *Left*: SR curves of FFN in layer 6 under varying LRs; More layers/modules can be found in Appendix A with similar trends. b) *Right*: SR-SJE relationships across training steps.

**SR-SJE Dynamics Driving Singularities**. We delve further into the cyclic interaction between Stable Jacobian Energy (SJE) and Stable Rank (SR), as visualized in Fig. 2(b). Intuitively, the SJE values stay less than 0.4 during the warm-up stage, indicating $< 40\%$ Jacobian energy is projected on the singular vectors within the stable ranks. As learning proceeds, we witness a constant upward trend of SJEs that reaches approximately 0.8 in late training stages, implying that around 80% of Jacobian energy is sprayed on the dominating singular directions, which is rather significant considering the stable ranks are even smaller than the warm-up stage. The SJE's increasing trend is complementary to the declining tendency of SR curves. As network singularity intensifies and dominant singular vectors gain prominence in the weight matrices, Jacobian matrices of high SJE will enhance the singular trend by further pushing parametric updates along those directions of significant singular values. This cycle of escalating singularities, captured by the SR-SJE relationship, underscores *the curse of singularities*, which ultimately leads to unstable losses and the breakdown of training stability.

**Training Breakdown.** Then, we investigate the network's behavior leading up to training instabilities, focusing on how escalating singularity drives loss explosions and training collapse. Fig. 3 (top row, blue) shows a case of loss explosion with an LR of $5 \times 10^{-4}$, while the bottom row (red)

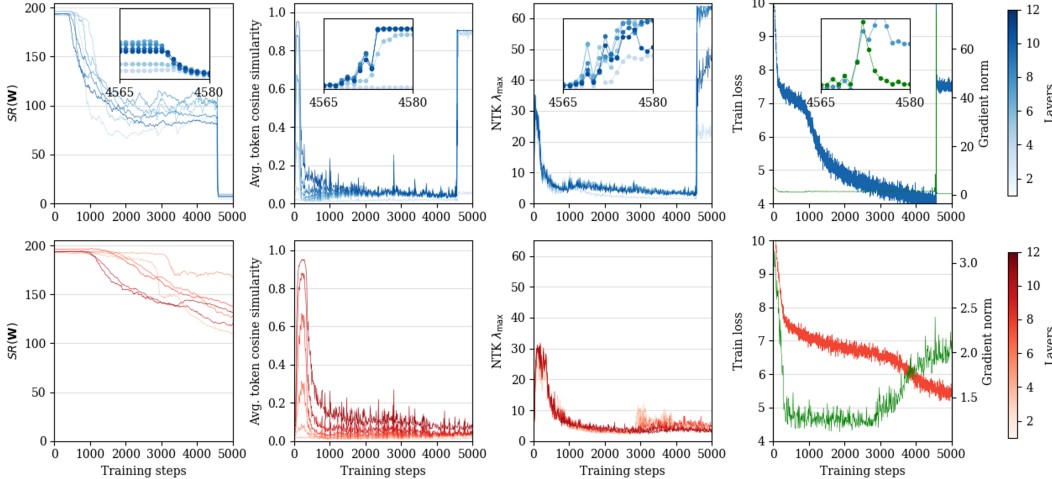

Figure 3: The top four panels depict the evolution of NTK-$\lambda_{\max}$ (for measuring gradient alignment), average token cosine similarity (between tokens from different sequences), Stable Rank(SR), training loss, and gradient norm during a case of loss explosion with a learning rate of $5 \times 10^{-4}$. The bottom four panels illustrate the same metrics during normal training with a learning rate of $1 \times 10^{-4}$.

represents normal training with an LR of $1 \times 10^{-4}$. The first column highlights an unusual plunge in the SR for the blue case, in contrast to the steady decrease in the red case. This sharp rise in parametric singularities leads to representations with overly high similarities in the feature space as shown in the second column of Fig. 3. The average cosine similarity between tokens reaches an unusually high 0.6 in shallow layers, climbing to an extreme 0.9 in deeper layers for the blue case, compared to typical values around 0.1 in the red one. Such similarity patterns suggest that network representations become rank-deficient and fail to retain sample-wise distinguishability.

This escalation of singularity, driven by the vicious cycle between Stable Jacobian Energy (SJE) and SR, culminates in severe rank deficiency and heightened gradient alignment. We quantify this alignment using the principal eigenvalue of a modified Neural Tangent Kernel (NTK) matrix (Jacot et al., 2018), where each element captures the dot product of normalized gradients between pairs of data points. The sharp rise of NTK $\lambda_{\max}$, as shown in the third column of Fig. 3 for the blue case, indicates that most gradients are increasingly aligning in the same direction, signaling the model's diminishing ability to learn diverse patterns from different data. Ultimately, these metrics reach critical values: token cosine similarities converge to 1.0 across all layers, and NTK $\lambda_{\max}$ hits its upper bound, meaning the gradients for all data points are fully aligned. This marks the complete collapse of the model's ability to differentiate between data points and capture distinct, nuanced patterns. The training loss escalates sharply and remains irreversibly high, signaling the breakdown of the learning process and the model's failure to recover.

## 3 METHOD

To dispel the curse of singularities, we propose a training strategy named Parametric Singularity Smoothing (PSS) to *detect* approaching instability and *prevent* its occurrence by smoothing the singular spectra of weight matrices when necessary. PSS is an easy-to-implement, efficient, and robust method capable of both *rescuing* network from suffering instabilities and *restoring* the trainability even after the loss explosions.

**Detection**. The curse of singularities is observed to constantly befall with gradients of significantly increasing magnitudes. Thus, we adopt gradient norms that can be fetched off-the-shelf during backpropagation as an effective indicator to launch the smoothing operations. Specifically, we track the ratio of the current step's gradient norms to the average gradient norms of all training steps, denoted as $\mu_{t+1} = \frac{\|\mathbf{g}^{t+1}\|_F}{\|\mathbf{g}^t_{avg}\|_F}$, where $\mathbf{g}^{t+1}$ is the gradient at step $t+1$ of the model, and $\mathbf{g}^t_{avg}$ is the historical average gradient over the previous $t$ steps, calculated using an exponentially weighted moving average: $\mathbf{g}^t_{avg} = (1-\alpha)\mathbf{g}^{t-1}_{avg} + \alpha\mathbf{g}^t$. Here $\alpha$ is the smoothing coefficient, typically $0 <$

$\alpha \leq 1$. If it is spotted that $\mu_{t+1} \geq \tau$ where $\tau$ is a threshold (we empirically set $\tau$ to 2.5 throughout the experiments), the model is considered to be stepping into the zone of potential instabilities, and hence, the smoothing operation is required to mediate the parametric singularities. We emphasize that the choice of $\tau$ is robust: a lower $\tau$ may cause false positives but without affecting normal training, while a higher $\tau$ may miss instabilities, but PSS can still rescue and restore stable training even after instability has occurred.

**Protection.**   Upon detecting instability, we mitigate the singularity of the parameter matrix $\mathbf{W}$ in each linear module by smoothing the singular spectrum. To reduce computational cost, we decompose only the dominant singular values and their corresponding singular vectors, limited in number by the SR, reduce the dominant singular values, and reparameterize the matrix using the corresponding singular vectors. Specifically, we first compute the largest singular value $\sigma_1$ using the Power Iteration Method (Golub & Van Loan, 2013), then calculate the $\text{SR}(\mathbf{W})$. With this, we apply Dominant Direction Decomposition (DDD) (Halko et al., 2009) to decompose the dominant part of the parameter matrix:

$$\mathbf{W}_{dominant} = \sum_{i=1}^{\lfloor \text{SR}(\mathbf{W}) \rfloor} \sigma_i \mathbf{u}_i \mathbf{v}_i^\top,$$

where $\boldsymbol{\sigma} = \{\sigma_i\}_{i=1}^{\lfloor \text{SR}(\mathbf{W}) \rfloor}$ denotes dominant singular values of the parameter matrix $\mathbf{W}$.

We apply a smooth function $f_{\text{smooth}}$ to $\boldsymbol{\sigma}$ to get the smoothed singular values

$$\boldsymbol{\sigma}^* = \{\sigma_i^*\}_{i=1}^{\lfloor \text{SR}(\mathbf{W}) \rfloor} = f_{\text{smooth}}(\boldsymbol{\sigma}),$$

and we then reparameterize the dominant part $\mathbf{W}_{dominant}$ without altering the singular vectors:

$$\mathbf{W}_{dominant}^* = \sum_{i=1}^{\lfloor \text{SR}(\mathbf{W}) \rfloor} \sigma_i^* \mathbf{u}_i \mathbf{v}_i^\top,$$

resulting in the smoothed parameter matrix:

$$\mathbf{W}^* = \mathbf{W}_{dominant}^* + \mathbf{W} - \mathbf{W}_{dominant}.$$

Given that the dominant singular vectors of the parameter matrix capture essential features from the data, the goal during smoothing is to avoid overly diminishing their magnitude. We select $\sigma_{\lfloor \text{SR}(\mathbf{W}) \rfloor + 1}$ as a threshold, ensuring that all dominant singular values remain above this level.

The choice of $f_{\text{smooth}}$ remains flexible and robust. Sub-linear functions such as Logarithmic with Scaling, Softplus, Softmax, or smoothing operations like convolution that maintain the relative order of dominant singular values are effective. Additionally, we show that an aggressive clipping-based method, where all dominant singular values $\sigma_i$ are reduced to $\sigma_{\lfloor \text{SR}(\mathbf{W}) \rfloor + 1}$, also demonstrates effectiveness. Ultimately, the objective is to increase the SR of weight matrices, with the specific choice of smoothing function adaptable to the task.

By integrating detection and protection, PSS enables early intervention to prevent impending instability, stabilizes the parameter matrix under large LRs, and can rescue and restore stability even after instability has occurred. This leads to robust and reliable training dynamics.

## 4 EXPERIMENTS

### 4.1 EXPERIMENTAL SETUP

**Model configuration.**   We conduct experiments on both the BERT (Devlin et al., 2019) and GPT-2 (Radford et al., 2019) models. We evaluate the effectiveness and robustness of PSS method by exploring its performance across different model scales and LRs. We present the main experimental results on BERT-base (110M parameters) and GPT-2-Medium (345M). We validate our method on larger models, including BERT-large (340M), GPT-2-Large (774M) and GPT-2-XL (1.5B), and details are in the appendix C.2 For the Masked Language Modeling (MLM) task, we train BERT from scratch using the Wikitext (Merity et al., 2016) dataset, and for the Causal Language Modeling (CLM) task we train GPT-2 using the mixed dataset of Amazon-review and the OpenWeb-Text (Gokaslan & Cohen, 2019). Both are trained using a LR schedule with warm-up followed by

decay. We present experimental results across a wide range of LRs for each model scale, spanning from rates that enable stable convergence to those that lead to instability. More detailed configurations are provided in the appendix C.1. Our approach is compared against two stabilization methods, including Gradient Clipping (GC) and Orthogonal Regularization (OR) (Brock et al., 2017; Brock, 2018).

**Evaluation metrics.** To assess the effectiveness of the stabilization method, we track the frequency of loss explosions across multiple trials under the same configurations following Zhai et al. (2023). While maintaining stable training, we analyze the expansion of the LR range for which stable training occurs by presenting convergence performance under various LR settings. We evaluate the impact of each method on the overall model training efficiency duration by evaluating the required floating-point computations(FLOPs) (Rasley et al., 2020) and time overhead.

## 4.2 Main Results

| Method \ Model | BERT-base | | | GPT-2-Medium | | |
|---|---|---|---|---|---|---|
| | lr=1e-4 | lr=2e-4 | lr=4e-4 | lr=5e-4 | lr=1e-3 | lr=2e-3 |
| Naive | $0/3_{3.0\pm0.2}$ | $3/3_{7.5\pm0.1}$ | $3/3_{7.5\pm0.1}$ | $0/3_{3.8\pm0.1}$ | $3/3_{7.4\pm0.1}$ | $3/3_{7.5\pm0.1}$ |
| GC | $0/3_{2.9\pm0.1}$ | $0/3_{2.7\pm0.1}$ | $3/3_{7.4\pm0.1}$ | $0/3_{3.7\pm0.1}$ | $0/3_{3.9\pm0.1}$ | $0/3_{5.0\pm0.1}$ |
| OR | $0/3_{3.1\pm0.1}$ | $1/3_{2.9\pm0.1}$ | $3/3_{7.4\pm0.1}$ | $0/3_{3.8\pm0.1}$ | $3/3_{7.5\pm0.1}$ | $3/3_{7.4\pm0.1}$ |
| PSS | $0/3_{3.0\pm0.1}$ | $0/3_{2.8\pm0.1}$ | $0/3_{2.6\pm0.1}$ | $0/3_{3.8\pm0.1}$ | $0/3_{4.0\pm0.1}$ | $0/3_{4.4\pm0.1}$ |

Table 1: Results of stability test on BERT-base and GPT-2-Medium models. In the context of the x/y (z) entry, y denotes the number of evaluations performed for the considered method, x represents instances where the model exhibited instability, and z signifies the perplexity of the model at the conclusion of the training process.

**Stability.** To evaluate the effectiveness of the stabilization method, we explore different LRs from stable convergence to loss explosion. For each fixed experimental configuration, we repeat the experiments three times using different fixed random seeds and measure the frequency of loss explosions to assess the reliability of the method, ensuring that the stabilization approach can definitively resolve instability issues.

Table 1 presents the comparison result, indicating that the GC and OR methods can only handle scenarios with small LRs, but at the cost of reducing the convergence speed and degrading the final performance. At moderately higher LRs, they can only slightly alleviate instability issues but cannot reliably and definitively stable the model training, and at even larger LRs, they completely fail. In contrast, our method ensures stable training across all configurations without compromising the model's convergence performance.

Fig. 4(a) illustrates the test loss curves for different methods on the BERT-base model at LRs of 1e-4 and 4e-4. Our method effectively detect two instances of loss explosion and quickly stabilize the training, with only minor, short-lived spikes in loss. Furthermore, it ensures stability without significant additional training steps to recover to the pre-explosion loss levels.

We investigate the usable LR range once stable training is ensured. Fig. 4(b) presents the final test loss across different LRs for the BERT-base and GPT-2-Medium models. On the BERT-base model, the GC and OR methods expand the maximum LR from 1e-4 to 2e-4, a twofold increase, while our method pushed it to 2e-3, achieving a 10-fold improvement. Similarly, on the GPT-2-Medium model, our method extends the LR range by by 10 times, outperforming other approaches. We further validate our method on larger models, including BERT-large, GPT-2-Large and GPT-2-XL, finding that PSS expand the LR range by up to 5 times without compromising convergence. Details are in the appendix C.2. This broader range allows model engineers to set LR hyper-parameters more flexibly without the risk of instability, enabling the use of larger LRs and reducing the complexity and time required for hyper-parameter tuning.

Another potential benefit of PSS expanding the LR range is improved performance at convergence. Typically, LR and final performance exhibit a U-shaped relationship. However, the training process may suffer from instability when the LR is near the bottom of the curve. As shown in Fig. 4(b),

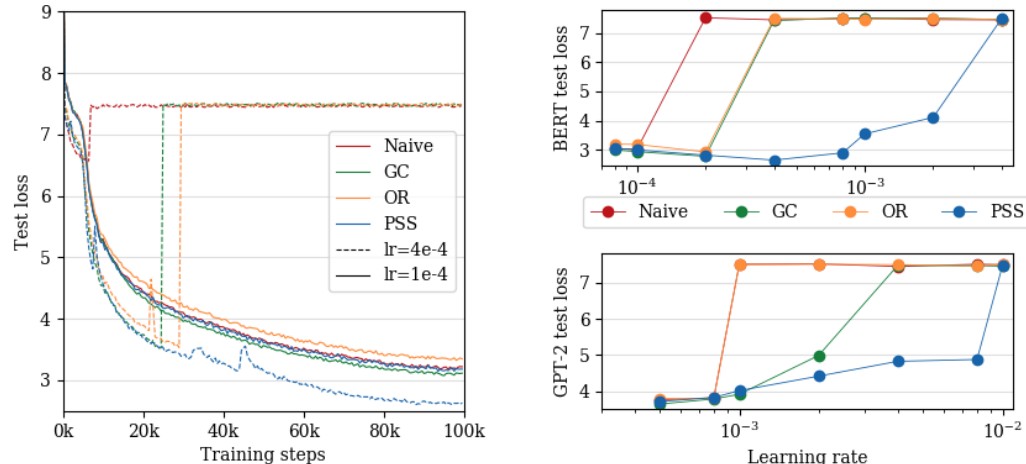

Figure 4: a) *Left*: Test loss curves of each stabilization method on two LRs: 1e-4 and 4e-4 in BERT-base model. b) *right*: The varying LRs and corresponding convergence performance of each stabilization method in BERT-base and GPT2-Medium models.

| Method | CPU+GPU cost @ 1 step | | CPU+GPU cost @ 100k step | |
|--------|-----------|------|-----------|------|
| | Fp computations | Time | Fp computations | Time |
| Naive | 9.33 TFLOPs | $781 \pm 10$ ms | 933 PFLOPs | $9.15 \pm 0.08$ h |
| GC | 9.33 TFLOPs | $784 \pm 12$ ms | 936 PFLOPs | $9.14 \pm 0.10$ h |
| OR | 11.46 TFLOPs | $864 \pm 15$ ms | 1146 PFLOPs | $10.04 \pm 0.10$ h |
| PSS | 9.36 TFLOPs | $1979 \pm 25$ ms | 936 PFLOPs | $9.17 \pm 0.09$ h |

Table 2: Computational cost comparison of different methods in terms of floating-point (Fp) computations and execution time for both a single step and 100k steps.

under the BERT-base configuration, our method's expanded LR range enables it to achieve a lower test loss, which other methods fail to stabilize.

**Computation overhead.** For a parameter matrix $W \in \mathbb{R}^{m \times n}$, in the detection policy, we use gradient norms as early indicators of instability, with a negligible computational cost of $O(mn)$. In the protection policy, the computational complexity of DDD on the matrix with $\lfloor SR(\mathbf{W}) \rfloor = k$ is $\mathcal{O}(mn \log k)$. For a feature matrix $X \in \mathbb{R}^{n \times b}$, where $n$ is the feature dimension and $b$ the batch size, the forward and backward pass complexity is $\mathcal{O}(mnb)$. In practice, since $b$ and $\log k$ are of the same order of magnitude, the cost of DDD is comparable to that of a forward-backward step. PSS's protection is triggered fewer than 10 times in most cases, and even in extreme LR settings, the activation frequency is less than 0.1% of the total steps, adding virtually little overhead to the training process.

We experimentally demonstrate the computational cost of our method in Table 2. In the "1 step" section, we show the cost of using the protection strategy when PSS detects instability. It can be seen that a single invocation of PSS incurs up to 2.40 times the CPU + GPU time of a single step in the naive baseline. However, since the protection policy is rarely triggered, over the full training process, PSS introduces nearly no additional overhead. In the BERT-base model, the additional computational overhead introduced by our method accounts for only 0.21% of the baseline training time. Moreover, compared to the regularization methods that similarly adjust singularity, PSS achieves a %8.5 improvement in overall training time.

### 4.3 ROBUSTNESS OF PSS

Our method demonstrates exceptional robustness, specifically in the following three aspects:

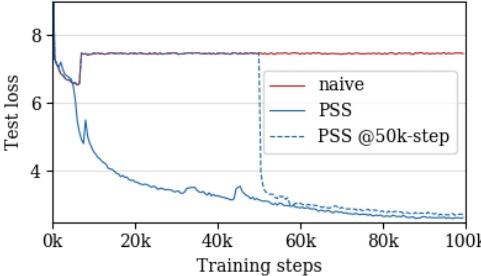 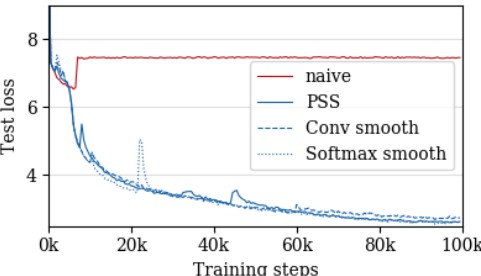

Figure 5: The experiment is conducted on the BERT-base model, trained with a learning rate of 4e-4. a) *Left*: The test loss curve in the naive baseline, which experiences instability, compared to the test loss curve when PSS is applied at different stages. b) *Right*: The test loss curve using different smoothing policies upon detecting instability.

**Rescue Robustness.** Fig. 5(a) highlights our method's effectiveness across different instability stages. When applied preemptively, it causes only brief, minor loss spikes while preventing future instability. If applied after divergence, it quickly halts gradient vanishing and restores normal loss descent. In contrast, prior methods fail in such cases, requiring trial-and-error tuning from saved checkpoints. This demonstrates the robustness of our detection metric: false positives do not disrupt training, and even after full divergence, the method reliably restores stability.

**Smoothing Policy Robustness.** Fig. 5 demonstrates the effectiveness of various smoothing policies, indicating that all are effective. Details of each policy are provided in the Appendix B.1. This suggests that as long as parametric singularity is reduced, the choice of smoothing policy becomes less critical, highlighting the robustness of the policy selection.

**Compatibility Robustness.** Our method imposes no specific requirements on model architecture, optimizer, or dataset, and is highly flexible and easy to integrate, requiring only minimal code changes. Furthermore, it is orthogonal to other stabilization methods, allowing for simultaneous use to further enhance training stability.

## 4.4 TRAINING DYNAMICS WITH OUR METHOD

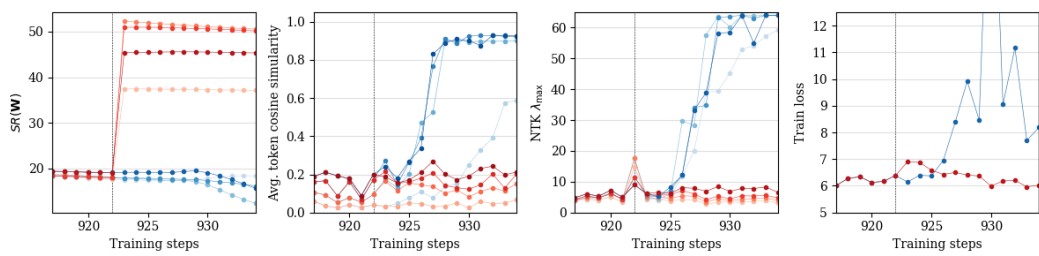

Figure 6: The effect of our method on the analytical metrics before, including SR, NTK-$\lambda_{max}$ and token-level cosine similarity. Each metric varies at the time our method is adapted

To better understand how our method stabilizes model training, Fig. 6 shows the changes in key metrics from the analysis section, including SR, NTK-$\lambda_{max}$ and token-level cosine similarity, after applying our method. When instability occurs, both NTK-$\lambda_{max}$ and token-level cosine similarity spike sharply, as noted in the analysis section, while the SR remains low. After applying our method, these metrics return to normal levels, and the SR increases, allowing the model to resume stable training.

## 5 RELATED WORK

**Large LRs.** Large LRs can offer significant benefits in deep learning. Jastrzebski et al. (2020) shows that large LRs improve the conditioning of the covariance of gradients, leading to flatter minima that generalize better (Lewkowycz et al., 2020; Lu et al., 2023; Huang et al., 2020; Leclerc & Madry, 2020). Additionally, Li et al. (2019) demonstrates from a feature learning perspective that small LRs cause models to first memorize low-noise, hard-to-fit patterns, resulting in poorer generalization. Smith (2017) and Smith & Topin (2019) show that cyclical large LRs can significantly accelerate model convergence. However, the instability caused by large LRs has been widely observed in empirical studies. Theoretical analyses often focus on sharpness bounds (Cohen et al., 2021), and Wang & Roberts (2023) explains this instability through landscape flattening and landscape shift. We establish a link between large LRs and parametric singularity, introducing the concept of the "curse of singularities" to explain the resulting instability.

**Training Stability.** Training instability has been widely studied from a theoretical perspective. Techniques like meticulous weight initialization (Glorot & Bengio, 2010; He et al., 2015; Mishkin & Matas, 2015; Hu et al., 2020) and normalization (Ba et al., 2016; Ioffe & Szegedy, 2015; Ulyanov et al., 2016) are critical for stabilizing neural networks. Additionally, research has examined the effects of activation non-linearity (Hu, 2016), skip connections (Szegedy et al., 2017), and model depth and width (Hanin, 2018; Yang & Schoenholz, 2018) on stability. In practice, gradient clipping (Allen-Zhu et al., 2019) is a common method for managing instability by capping gradient magnitudes. Adaptive methods, such as those proposed by Tang et al. (2023), improve on this by incorporating optimizers like Adagrad. Other approaches include selectively freezing parameters in ViT models (Chen et al., 2021) and using spectral normalization to prevent attention entropy collapse (Zhai et al., 2023), further stabilizing training. Despite the aforementioned stability methods, models still encounter instability under large LRs, and these methods prove ineffective once instability arises. Our PSS method significantly broadens the usable LR range and ensures reliable training.

## 6 CONCLUSION

This work tackles the challenge of training instability from large LRs by identifying parametric singularity as a key factor. Excessive singularity in weight matrices leads to rank deficiencies and loss explosions. We uncover a vicious cycle, the "curse of singularities", that worsens instability as training progresses, ultimately leading to rank-deficient representation and resulting in loss explosion. Our method smooths singular spectra, enabling stable use of large LRs. This foundation offers insights into managing singularities and stabilizing training. We show our approach's robustness across networks, extending LR ranges and reducing training times.

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

## A  DETAILED ANALYSIS

We conducted a detailed parametric singularity analysis on the BERT-base model. In Figs8, 9 and 10, we present the dynamic changes in the average stable rank of the Attention-head, Attention-Dense and FFN modules across different layers and learning rates. It can be observed that, as training progresses, the stable rank of various modules decreases across layers, with deeper layers showing consistently lower stable ranks compared to shallower ones. Additionally, regarding the learning rate, we observe that larger learning rates accelerate the growing singularity trends, and the parameter matrices tend to converge to a lower stable rank.

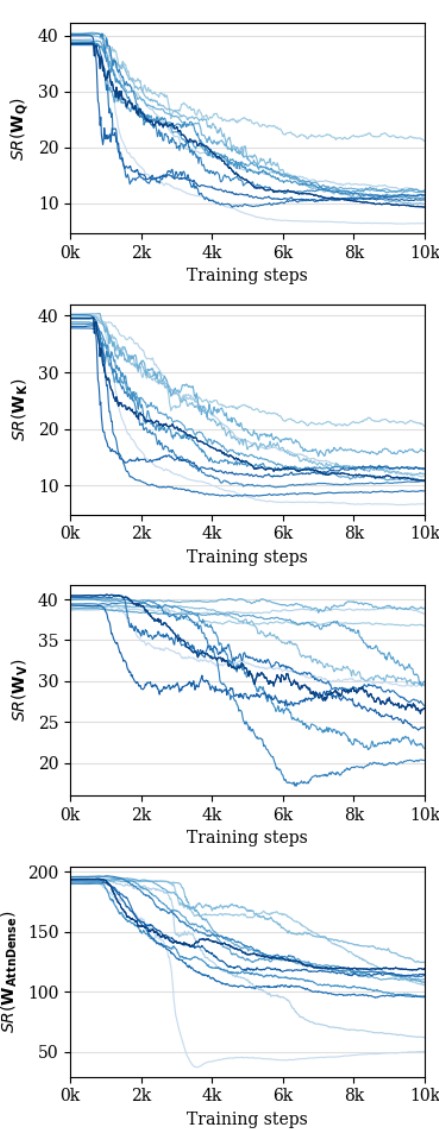

Figure 7: The dynamic changes in the average stable rank of the Attention-head modules across different layers and learning rates in BERT-base model.

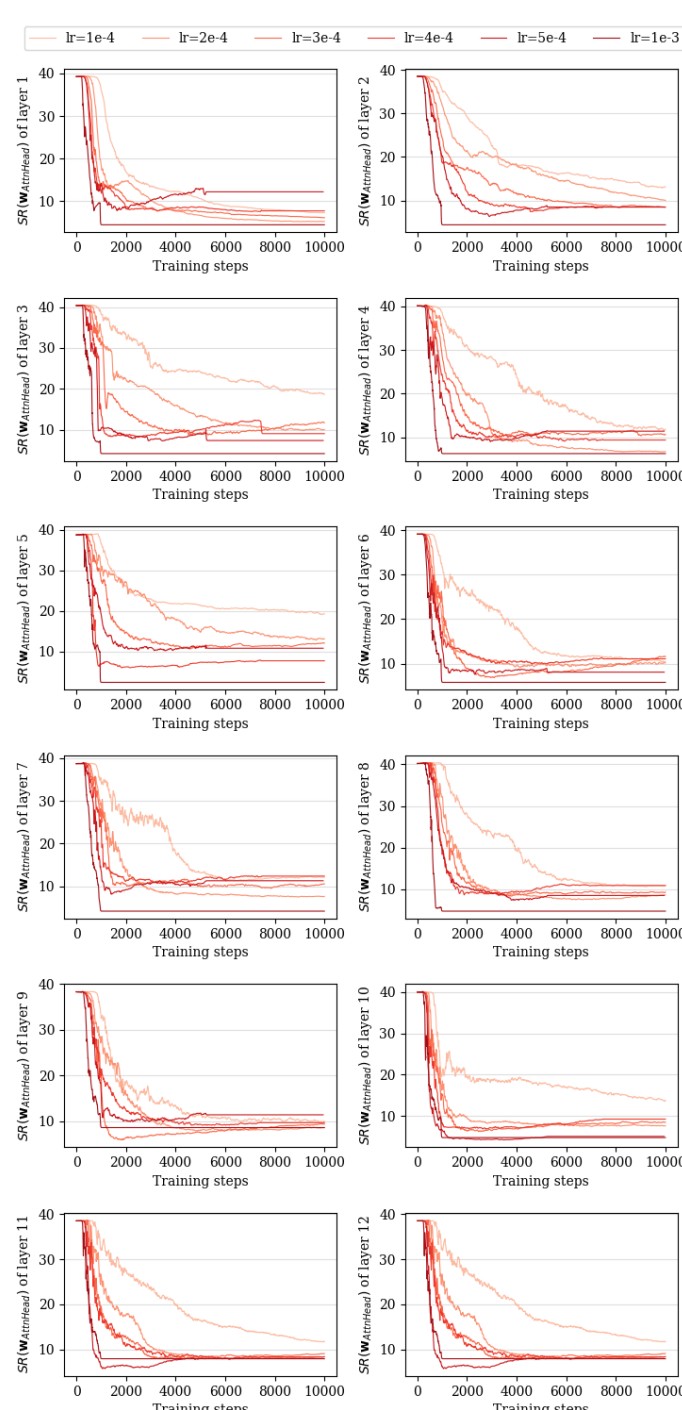

Figure 8: The dynamic changes in the average stable rank of the Attention-head modules across different layers and learning rates in BERT-base model.

# B METHOD DETAILS

## B.1 SMOOTHING POLICY

Given a weight matrix $\mathbf{W}$ and its dominant singular values $\boldsymbol{\sigma} = \{\sigma_i\}_{i=1}^{\lceil \mathrm{SR}(\mathbf{W}) \rceil}$, we introduce several smoothing functions below.

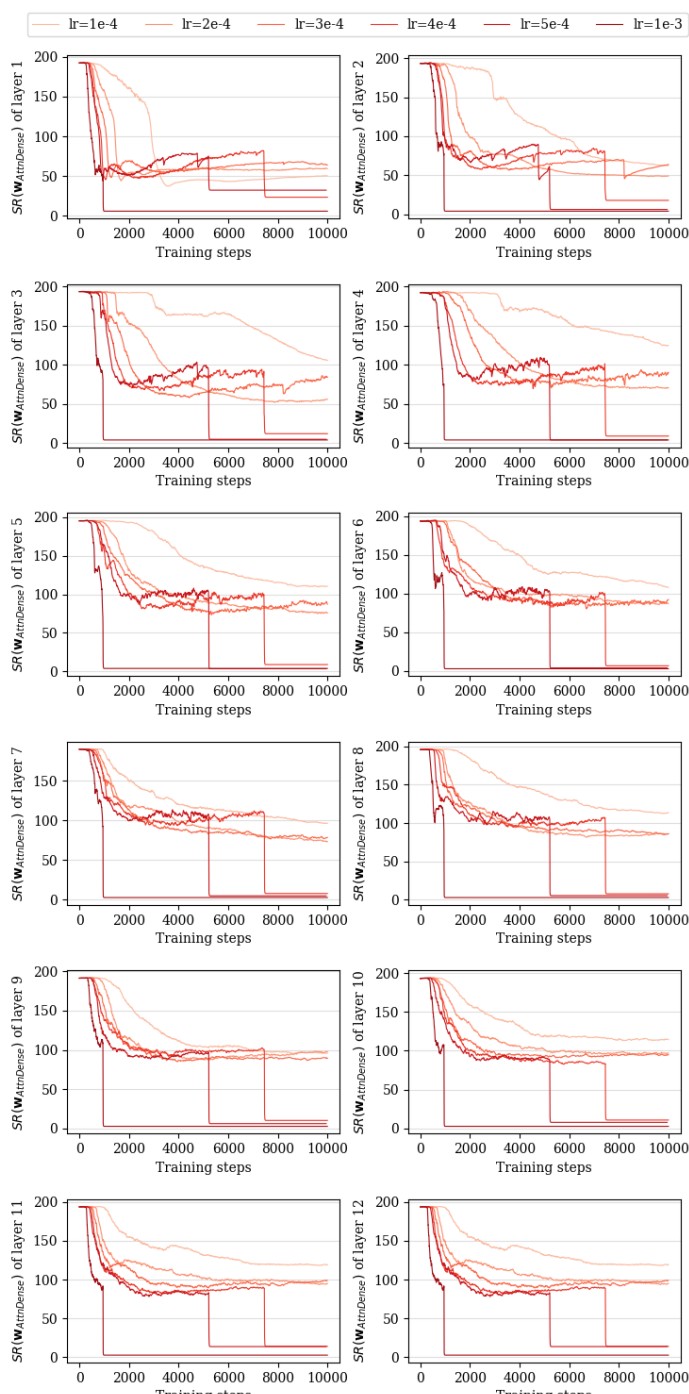

Figure 9: The dynamic changes in the average stable rank of the Attention-Dense modules across different layers and learning rates in BERT-base model.

**Convolution.** We apply a convolution kernel $\mathbf{k} = \{k_j\}_{j=-m}^{m}$ to the dominant singular values $\boldsymbol{\sigma}$ to smooth the transition between them. The smoothed singular values $\boldsymbol{\sigma}^{\text{smooth}} = \{\sigma_i^{\text{smooth}}\}$ are computed by convolving the kernel with the original singular values:

$$\sigma_i^{\text{smooth}} = \sum_{j=-m}^{m} k_j \cdot \sigma_{i+j}, \quad \forall i \in \{m+1, \dots, \lceil \text{SR}(\mathbf{W}) \rceil - m\}$$

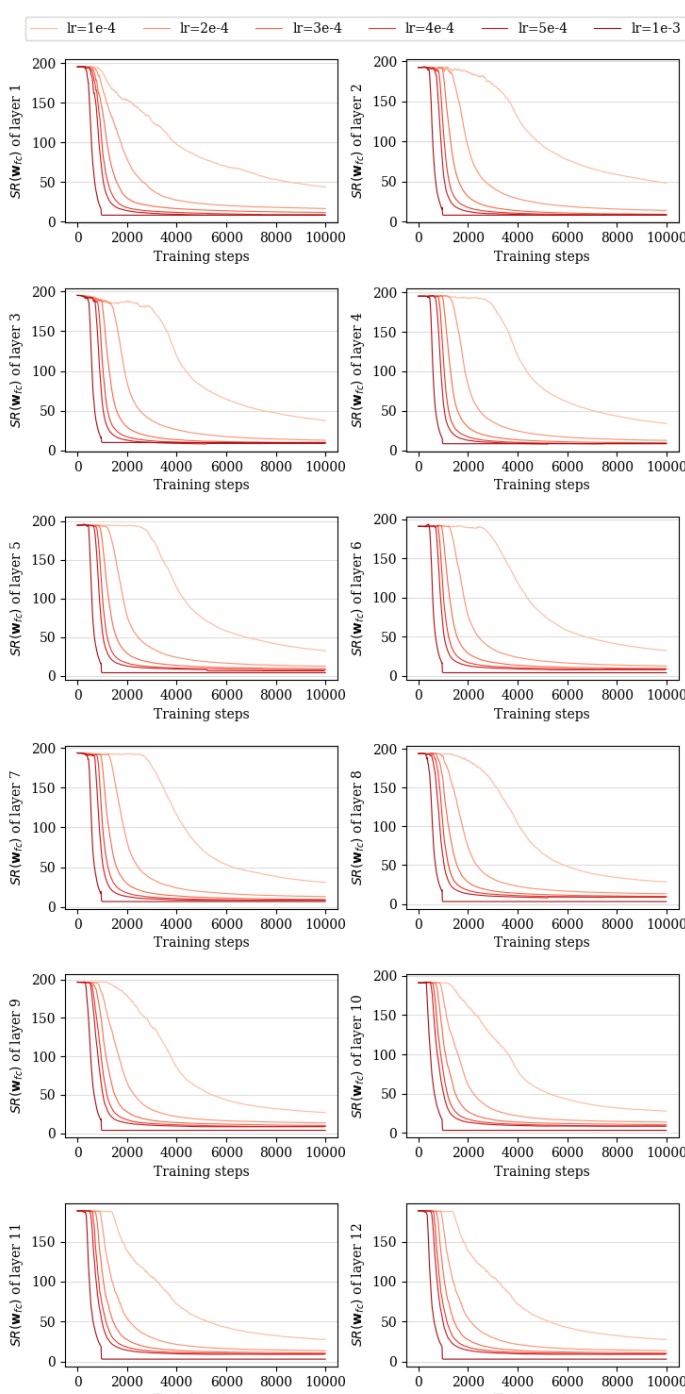

Figure 10: The dynamic changes in the average stable rank of the FFN modules across different layers and learning rates in BERT-base model.

where $\sigma_{i+j}$ refers to neighboring singular values, and the kernel $\mathbf{k}$ is typically normalized to ensure that the sum of the kernel elements equals 1:

$$\sum_{j=-m}^{m} k_j = 1.$$

This convolution operation smooths the singular value spectrum by averaging over a local neighborhood of values. It reduces abrupt changes between adjacent singular values while preserving

their relative order, ensuring that the most dominant singular values retain their influence in the parametric space.

**Softmax.** We apply the Softmax function to the dominant singular values $\boldsymbol{\sigma} = \{\sigma_i\}_{i=1}^{\lceil \text{SR}(\mathbf{W}) \rceil}$ to ensure that the singular values are normalized while retaining their relative importance. The Softmax-smoothed singular values $\boldsymbol{\sigma}^{\text{softmax}} = \{\sigma_i^{\text{softmax}}\}$ are computed as follows:

$$\sigma_i^{\text{softmax}} = \frac{\exp(\beta \cdot \sigma_i)}{\sum_{j=1}^{\lceil \text{SR}(\mathbf{W}) \rceil} \exp(\beta \cdot \sigma_j)}$$

where $\beta$ is a scaling parameter that controls the sharpness of the distribution. A higher $\beta$ places more emphasis on the larger singular values, while a lower $\beta$ spreads the emphasis more evenly across all singular values.

The Softmax function smooths the singular values by normalizing them in a way that maintains the hierarchy of dominance, but it prevents any one singular value from dominating excessively. This ensures that the singular value distribution remains balanced and well-behaved during the smoothing process.

## C  EXPERIMENT DEATILS

### C.1  MODEL CONFIGURATION DETAILS

The detailed training configuration of BERT and GPT-2 in shown in Table 3 and Table 4.

| Configurations | BERT-base | BERT-large |
|---|---|---|
| Hidden activation | GELU | |
| Max position embeddings | 512 | |
| Tokenizer | RoBERTa-base | |
| Vocabulary size | 50265 | |
| Sequence length | 128 | |
| Layernorm $\epsilon$ | 1e-12 | |
| Dropout probability | 0.1 | |
| Optimizer | AdamW | |
| AdamW weight decay | 0.01 | |
| AdamW $(\beta_1, \beta_2)$ | (0.9, 0.999) | |
| AdamW $\epsilon$ | 0.01 | |
| LR warm-up steps | 1000 | |
| LR schedular | Cosine | |
| Hidden size | 768 | 1024 |
| Intermediate size | $4 \times$ Hidden size | |
| Hidden Layers | 12 | 24 |
| Num. attention heads | 12 | 16 |
| Batch size | 64 | 32 |

Table 3: Configuration of BERT training.

### C.2  EXPERIMENT RESULTS ON LARGER MODELS

We validated the effectiveness of our method on larger models, including BERT-large (340M), GPT-2-Large (774M), and GPT-2-XL (1.5B). For these larger models, we primarily compared with the naive baseline to demonstrate our method's ability to stabilize training. In all three models, we successfully used large learning rates, which would cause instability in the naive baseline, to achieve stable training.

| Configurations | GPT-2-medium | GPT-2-large | GPT-2-XL |
|---|---|---|---|
| Hidden activation | | GELU | |
| Max position embeddings | | 512 | |
| Tokenizer | | r50-k | |
| Vocabulary size | | 50257 | |
| Sequence length | | 256 | |
| Layernorm $\epsilon$ | | 1e-5 | |
| Dropout probability | | 0.5 | |
| Optimizer | | AdamW | |
| AdamW weight decay | | 0.1 | |
| AdamW $(\beta_1, \beta_2)$ | | (0.9, 0.995) | |
| AdamW $\epsilon$ | | 0.01 | |
| LR warm-up steps | | 1000 | |
| LR schedular | | Linear | |
| Hidden size | 1024 | 1280 | 1600 |
| Intermediate size | | $4 \times$ Hidden size | |
| Hidden Layers | 24 | 36 | 48 |
| Num. attention heads | 16 | 20 | 25 |
| Batch size | 64 | 32 | 16 |

Table 4: Configuration of GPT-2 training.

| Model / Method | BERT-Large | | GPT-2-Large | | GPT-2-XL | |
|---|---|---|---|---|---|---|
| | lr=5e-5 | lr=1e-4 | 5e-5 | 1e-4 | 1e-5 | 5e-5 |
| Naive | $0/2_{2.4\pm0.1}$ | $2/2_-$ | $0/1_{3.9}$ | $1/1_-$ | $0/1_{3.7}$ | $1/1_-$ |
| PSS | $0/2_{2.4\pm0.1}$ | $0/2_{2.7\pm0.1}$ | $0/1_{4.0}$ | $0/1_{4.2}$ | $0/1_{3.8}$ | $0/1_{3.9}$ |

Table 5: Results on BERT-Large, GPT-2-Large and GPT-2-XL