# OpenReview forum: "Large Learning Rates without the Agonizing Pain: Dispelling the Curse of Singularities in Deep Neural Networks"
_ICLR.cc/2025/Conference — ICLR 2025 Conference Withdrawn Submission_

### Official Review · Reviewer_G67M · 2024-10-28

**Soundness:** 3
**Presentation:** 3
**Contribution:** 2
**Rating:** 5
**Confidence:** 2

**Summary:**

The article introduces Parametric Singularity Smoothing, a technique designed to stabilize neural network training with large learning rates by addressing singularities, or rank deficiencies, in weight matrices. Large learning rates often lead to instability and to explosions due to increase in parametric singularities in the network during training. PSS mitigates these issues by monitoring Stable Rank and Stable Jacobian Energy, which measure the concentration of singular values in weight matrices and the distribution of gradient energy, respectively. When SR drops or SJE rises sharply—indicating an impending loss explosion—PSS triggers a smoothing process, where dominant singular values are adjusted using techniques like convolution or softmax to prevent one direction from overpowering others. This controlled smoothing process rebalances the weight matrix, reducing the risk of instability without extensive hyperparameter tuning. Empirical results across  BERT and GPT-2, show that PSS can significantly expand the usable learning rate range without exploding with minimal overhead.

**Strengths:**

S1. The article empirically identifies singularities as a primary cause of instability at high learning rates.

S2. The article demonstrates that PSS can broaden the usable learning rate range, enhancing training efficiency.

S3. PSS is simple to implement, with minimal computational overhead.

S4. The article introduces novel quantitative stability metrics, providing clear ways to assess and compare singularity levels.

**Weaknesses:**

W1. While shown empirically, the link between singularities and instability isn't addressed theoretically.

W2. PSS requires a threshold and smoothing method. The article demonstrates the robustness of smoothing methods but not the threshold. Isn't the fine-tuning of the threshold similar to the fine-tuning of LR?

W3. The article shows limited comparisons with other stabilization methods mentioned in the related work section.

W4. Some figures lack clarity (e.g., unclear indicators in Figures 3 and 6).

W5. Effectiveness in downstream tasks isn’t fully explored, raising questions about potential drawbacks during training.

W6. Besides preventing explosions, results don’t clearly show faster or improved convergence at large learning rates.

W7. The Protection section notations are hard to follow.

Overall, I am not convinced that the paper's contribution is of great significance. The article seems to solve a marginal problem. The other regularization methods provide comparable results in preventing explosions, and the current work is not convincing that explosions are a frequent problem with a scheduler.

**Questions:**

See weaknesses.

Beside empirical results, is there a theoretical way to show the link between singularities and instability?

---

> ### Author Response · Authors · 2024-11-27
> **Withdrawal of Submission**
>
> Thank you for your thoughtful feedback and valuable insights on our submission. After careful consideration, we have decided to withdraw the paper from consideration at this time to refine and improve our work further.
>
> Thank you once again for your understanding.

---

### Official Review · Reviewer_B3yp · 2024-10-28

**Soundness:** 4
**Presentation:** 3
**Contribution:** 2
**Rating:** 3
**Confidence:** 4

**Summary:**

The paper proposes a new lightweight method to alleviate the challenges linked with training with large weight decay. To prevent training instabilities that can lead to a complete training collapse, the authors propose a method which first detects the potential emergence of instability by measuring the relative changes in gradient norm and then smoothens the singular value spectrum by smoothing the dominant singular values. An approximate, fast partial SVD alternative is used instead of the full SVD to prevent computation overhead. The authors evaluate the effectiveness of the method on several language datasets and LLMs. They show that the method enables the use of significantly larger learning rates than vanilla SGD or Adam at very small computational overhead. This stabilizes the training and allows for better generalization performance, while easing the practitioners of tedious learning rate tuning.

The authors motivate this method through the occurrence of training singularities in which the stable rank, NTK leading eigenvalue and other metrics significantly change and lead to irreversible loss jump. This is further complemented with the introduction of stable jacobian energy, a quantity that measures how much of the gradient update operates in the singular space corresponding to the dominant singular values. If this quantity is too large, it is an indication of a vicious cycle where the dominant singular values get even more reinforced and the tail dies irrevocably.

**Strengths:**

- S1: The main strength of the paper is that the proposed method is lightweight, neat and, according to the evidence from the paper, effective. It can be easily implemented and added to anyone's training routine.

- S2: The paper is well-written and easy to read. It also makes for an interesting read.

- S3: The method was evaluated on a reasonable representation of the modern language models, providing fairly robust evidence for its effectiveness *in this scope*.

**Weaknesses:**

- W1: The authors severely under-compare and under-refer an alternative approach of dealing with learning rate -- automatic learning rate setters. There are many papers that propose automatic learning rate computational schemes that run without manual interventions and promise the same fruit as this work -- large learning rates that however still lead to stable training dynamics (see for instance [1-6]). Some of these papers also mention that the resulting average learning rate is much higher than a stable learning rate using vanilla optimizers. Most of the papers don't come with too prohibitive computational overhead. I would expect the paper to compare against the most competitive methods from this area and discuss them in related work.

- W2: The work lacks some ablation studies. In particular, the paper should investigate, whether some similar, but less principled operation on singular values would still give the same improvements or not. Examples of simpler ideas -- adding random noise to the matirx as a mean of smoothening its spectrum, setting just the first singular value to the value of the second one, or using hard (thresholded) rank instead of the stable rank.

- W3: There is a rather weak link between the "theoretical" motivation and the method itself. In particular, the authors just smoothen the singular values that are *already dominant*, thus not fixing the imbalance between large and small singular values, nor the number of the large singular values. For instance, for the convolution smoothing, the sum of dominant singular values stays the same, not balancing the large and small values at all. In fact, the fact that the method can recover good training even after the loss jump has happened suggests that the reason why the method works has nothing to do with balancing the singular values. Note that since we change only the top singular values, the mismatch between large and small singular values remains. This suggests that it might be important to re-weight the dominant directions, rather than balance them with the weak directions. To elaborate on this even further, left plot in Figure 6 even suggests that the loss explosion might not be caused by singularities at all. The effective rank in the collapsed solution stays constant for the steps of collapse, suggesting the loss might be exploding for completely different reasons.

- W4: The introduction of the stable jacobian energy (SJE) seems pointless in this paper. It is not used at all in connection with the proposed method (in fact I expected it to be used as the indicator to launch the mechanism, instead of gradient norm. That would fit the paper's motivation much better). Moreover, it does not serve as a convincing motivation behind the emergence of singularities, because the paper doesn't provide enough empirical or theoretical evidence for it. The only piece of evidence is Fig. 2b, but that only shows some correlation that provides very little insight in what is happening at the time of the loss explosions. I would suggest the authors to fully drop this aspect, or to provide much more discussion on it.

- W5: I disagree with the authors that the method is robust w.r.t. the choice of the hyperparameter $\tau$. If you pick too small $\tau,$ you might evoke the mechanism too often, resulting in a significant slowdown (maybe even non-convergence). If the $\tau$ is too big, you might need manual intervention because the smoothing might not be evoked even after the loss jump. Moreover, the hyperparameter is also dependent on both the exponential averaging coefficient $\alpha,$ as well as the learning rate itself. Thus, it might be quite tricky to choose the correct $\tau,$ which undermines the original motivation of this method to avoid hyperparameter optimization.

- W6: In the paper, the authors claim that the gradient clipping method is more computationally heavy (line 357) than PSS, but this disagrees with the evidence and also doesn't make sense since gradient clipping is extremely cheap. Moreover, the gradient clipping has only been demonstrated to perform worse for one particular setup. It would be nice to see a bit more convincing evidence that PSS significantly outperforms gradient clipping.

- W7: I don't think the method really allows to avoid learning rate tuning. I think one still needs to perform some learning rate tuning. This method just allows one to use higher values of lr safely.

[1] Jin, Yuchen, et al. "Autolrs: Automatic learning-rate schedule by bayesian optimization on the fly." arXiv preprint arXiv:2105.10762 (2021).

[2] Merrouchi, Mohamed, et al. "AutoLrOpt: An Efficient Optimizer Using Automatic Setting of Learning Rate for Deep Neural Networks." IEEE Access 12 (2024): 83154-83168.

[3] Yedida, Rahul, and Snehanshu Saha. "A novel adaptive learning rate scheduler for deep neural networks." arXiv preprint arXiv:1902.07399 (2019).

[4] Li, Yong, et al. "The improved training algorithm of back propagation neural network with self-adaptive learning rate." 2009 international conference on computational intelligence and natural computing. Vol. 1. IEEE, 2009.

[5] Subramanian, Shreyas, and Vignesh Ganapathiraman. "Zeroth Order GreedyLR: An Adaptive Learning Rate Scheduler for Deep Neural Network Training." 2023 IEEE 4th International Conference on Pattern Recognition and Machine Learning (PRML). IEEE, 2023.

[6] Wang, Kang, et al. "An automatic learning rate decay strategy for stochastic gradient descent optimization methods in neural networks." International Journal of Intelligent Systems 37.10 (2022): 7334-7355.

**Questions:**

- Q1: Have you measured the exact distribution of the singular values? Perhaps there is a structure that creates some biases of this method, such as the first singular value being significantly larger than all the others.

- Q2: Do all these claims apply to non-transformer architectures like the MLPs and CNNs? I have seen many unstable trainings of MLPs with large learning rate with a lot of spikes, but the loss almost always recovered automatically. Perhaps this phenomenon is transformer-specific?

- Q3: Do these catastrophic loss explosions happen even if you significantly decrease the weight decay? I notice you use weight decay of $0.1$ for GPT, which is a very large value. Perhaps with much smaller weight decay you could use much larger learning rates. It is well-known that the weight decay induces low rank bias.

- Q4: If you use the version of your method that significantly decreases the values of the singular values, such as the one where you cut all the singular values to be the smallest one of them, do you still get training speedup? The thing is your weight matrices now somewhat regrow, which might take some time.

- Q5: In line 264 you claim that the singularities and large gradients are closely connected. Do you have some theoretical explanation for this?

- Q6: The dominant direction decomposition method you use to estimate part of SVD is only approximate. Did you measure the precision of this method in practice?

- Q7: Which smoothing function do you use in most of your experiments? This has not been written anywhere in the paper, nor in the appendix (correct me if I am wrong).

- Q8: Why do all the exploded losses in figure 4 converge to the same number? Is this perhaps suggesting that all collapsed networks just output zeros? If it is so, then their rank is 0 and the soft rank is not a good measure for this scenario.

- Q9: What do the authors mean by the sentence: "Moreover, compared to the regularization methods that similarly adjust singularity, PSS
achieves a %8.5 improvement in overall training time." from line 427? And what about "prior methods fail in such cases" in line 452?

**Summary:**
Although the presented method is functional and neat, I don't think the paper is ready to be published at this point. The main reasons are: insufficient evidence that the method is needed and poor comparison with alternative approaches, poor link between theoretical motivation and the method itself, introduction of new hyperparameters that might not be as unimportant as the authors claim and unconvincing advantage over the methods the authors did compare against, such as the gradient clipping.

---

> ### Author Response · Authors · 2024-11-27
> **Withdrawal of Submission**
>
> Thank you for your thoughtful feedback and valuable insights on our submission. After careful consideration, we have decided to withdraw the paper from consideration at this time to refine and improve our work further.
>
> Thank you once again for your understanding.

---

### Official Review · Reviewer_iuLG · 2024-10-31

**Soundness:** 2
**Presentation:** 3
**Contribution:** 2
**Rating:** 6
**Confidence:** 3

**Summary:**

This work examines the challenges of training under large learning rates. They track two measures, the stable rank (SR) and the stable Jacobian energy (SJE), first per layer over the course of training training, and next in a certain layer for different learning rates over the course of training. They show that loss explosions under a large learning rate can be detected by SR. The second main part of the paper proposes a methodology they call PSS to first detect and then "protect" the training by "smoothing" the weight matrices.

**Strengths:**

* Originality: The smoothing procedure that produces $W^*$ is novel, to the best of my knowledge.
* Quality: The technical elements of the paper appear sound and well-executed.
* Clarity:  The paper defines its measures clearly, and the writing is generally strong.
* Significance: The smoothing procedure appears to enable a broader range of learning rates, representing a meaningful contribution.

**Weaknesses:**

* I think a main weakness of the paper is that Section 2 could be largely skipped in favor of moving directly to the method in Section 3. Figure 3 shows that several metrics, beyond SR, can indicate training breakdowns effectively.
* The protection policy laid out in Section 3 seems somewhat ad-hoc. Would you be comfortable describing how you arrived at it and perhaps failed attempts on the way?
* SR is integrated into the methodology, but SJE is not. Removing SJE may not impact the paper significantly, as its contribution is primarily limited to Figure 2b.

**Questions:**

* Equations (2) and (3) appear to depend on the input $x$, though this dependency is not shown explicitly in the notation. Additionally, in Section 2.2, when referring to "average SRs" in the caption of Figure 1, do you mean averaged over all $x$ values in the dataset? There are a few other instances where "average" is used, and these would benefit from clearer explanation, ideally with supporting mathematical expressions.
* In Line 158, the statement "while the observations are prevalent across networks and datasets" would benefit from a reference to the appendix if there is relevant material there.
* The color map tick marks in Figure 1 could follow the labeling used in Figure 3 (e.g., $2, 4, 6, 8, 10, 12$). The color schemes in Figures 2 and 3 could also be more accessible if they transitioned from cold to warm colors rather than staying within a single color family (blue or red).
* The Figure 3 caption has the order of NTK and SR switched, which should be corrected. Additionally, the fourth column in Figure 3 needs more context: what does the layer color represent since there's no breakdown by layer for training loss. Also, what is the significance of the green line? Finally, consistency in layer colormaps between the top and bottom rows would improve readability.
* The presentation of the smoothed parameter matrix in Line 297 could be strengthened by clarifying how this integrates into the overall training procedure.
* Throughout the paper, the term "naive baseline" is used, but the baseline itself is never defined. A brief description of what this baseline entails would add clarity.
* In the caption for Table 1, the format is noted as "$x/y(z)$", but the actual format is $x/y_{z1 \pm z2}$. Furthermore, the appearance of perplexity in Table 1 is unexpected and should be explained, especially regarding its relevance to the overall results and metrics discussed in the paper.

---

> ### Author Response · Authors · 2024-11-27
> **Withdrawal of Submission**
>
> Thank you for your thoughtful feedback and valuable insights on our submission. After careful consideration, we have decided to withdraw the paper from consideration at this time to refine and improve our work further.
>
> Thank you once again for your understanding.

---

### Official Review · Reviewer_SdTp · 2024-11-04

**Soundness:** 2
**Presentation:** 2
**Contribution:** 2
**Rating:** 5
**Confidence:** 2

**Summary:**

This work introduces a new technique for avoiding training instabilities during neural networks training. The main observation is that certain singularities are the culprits of training instabilities. Therefore, their method (called Parametric Singularity Smoothing) is devoted to avoiding such singular configurations and therefore prevent training instability.

**Strengths:**

-- The problem is very concrete, and anyone who has ever trained a neural network is acquainted with it. Therefore, their new approach could potentially have an impact in the field.
-- The numerical results seem encouraging.
-- Definitions 1 and 2, along with the fact that instability during training might be due to certain types of singularities in the parameter space is quite interesting. I think this is the main contribution of the article.

**Weaknesses:**

-- The paper seems excessively empirical. The theoretical aspect might be studied more. It might be very difficult perhaps, but here there is not even an attempt.
-- There is only an intuitively explanation about why the procedure introduced in the article should allow to avoid the singularities that cause instability. One might expect a bit more than that.

**Questions:**

-- Did the author try to go beyond the qualitative empirical evidence, and try to study the formation of singularities during training?
-- Is there any reasonable assumption that might guarantee that the PSS procedure would allow the user to avoid singularity formation?

---

> ### Author Response · Authors · 2024-11-27
> **Withdrawal of Submission**
>
> Thank you for your thoughtful feedback and valuable insights on our submission. After careful consideration, we have decided to withdraw the paper from consideration at this time to refine and improve our work further.
>
> Thank you once again for your understanding.

---

### Note · Authors · 2024-11-27

I have read and agree with the venue's withdrawal policy on behalf of myself and my co-authors.